# Repurposing Acetylcholinesterase Inhibitors for Leishmaniasis: Donepezil Hydrochloride and Related Compounds Against the American Tegumentary Form

**DOI:** 10.3390/antibiotics14121182

**Published:** 2025-11-21

**Authors:** Daniela E. Barraza, Emilse N. Araoz, María A. Occhionero, Daniela A. Gaspar, Eliana G. Guevara, María E. Vázquez, Brenda A. Zabala, Paola A. Barroso, Cecilia Pérez Brandán, Carlos J. Minahk, Leonardo Acuña

**Affiliations:** 1Unidad de Biotecnología y Protozoarios (UBIPRO), Instituto de Patología Experimental “Dr. Miguel Ángel Basombrío” (IPE), Consejo Nacional de Investigaciones Científicas y Técnicas (CONICET), Universidad Nacional de Salta, Salta A4408FVY, Argentina; barrazade86@gmail.com (D.E.B.); emilsearaoz48@gmail.com (E.N.A.); chely.mao94@gmail.com (M.A.O.); perezbrandan@gmail.com (C.P.B.); 2Instituto Superior de Investigaciones Biológicas (INSIBIO), Consejo Nacional de Investigaciones Científicas y Técnicas (CONICET), Universidad Nacional de Tucumán, San Miguel de Tucumán T4000ILI, Argentina

**Keywords:** leishmaniasis, drug repurposing, acetylcholinesterase inhibitors, donepezil hydrochloride, amphotericin B, drug synergy, neglected tropical diseases, antileishmanial agents

## Abstract

**Background/Objective:** American tegumentary leishmaniasis is a neglected tropical disease with limited therapeutic options characterized by high toxicity and poor tolerability. Drug repurpose offers a pragmatic strategy to accelerate the development of safer treatments. This study evaluated the antileishmanial activity of three clinically approved acetylcholinesterase (AChE) inhibitors—donepezil hydrochloride (DH), rivastigmine tartrate (RT), and galantamine hydrobromide (GH), tested individually and in combination with amphotericin B (AmpB) against *Leishmania* species relevant to tegumentary leishmaniasis. **Methods:** Antileishmanial activity was assessed against *Leishmania (Leishmania) amazonensis* promastigotes and intracellular amastigotes and *Leishmania (Viannia) braziliensis* promastigotes and axenic amastigotes. Cytotoxicity was evaluated in mammalian cell lines. The synergy with AmpB was analyzed at different proportions. Mechanistic studies included morphological analysis using light and scanning electron microscopy, flow cytometry, AChE activity assays, choline supplementation experiments, and membrane fluidity measurements. **Results:** All three AChE inhibitors demonstrated antileishmanial activity with selectivity indices > 1. DH emerged as the most promising candidate (IC_50_ = 16.82 μM against promastigotes; SI = 10.25), with superior potency compared to other repurposed drugs. Strong synergistic interactions with AmpB were observed for all inhibitors (χΣFIC ≤ 0.17), with DH-AmpB displaying the most robust synergy (χΣFIC = 0.09), reducing the IC _50_ of AmpB by nearly 90-fold. DH induced distinct morphological alterations and acted through non-cholinergic mechanisms. The DH-AmpB combination retained maximal efficacy against *L. (V.) braziliensis*, with enhanced activity against clinically relevant amastigotes. **Conclusions:** Repurposed AChE inhibitors, particularly donepezil hydrochloride, are highly promising therapeutic candidates for tegumentary leishmaniasis. The robust synergistic effect with amphotericin B, together with their favorable safety profiles and non-antimicrobial mechanisms, positions these drugs as viable partners in dose-sparing combination regimens that could improve treatment adherence and reduce toxicity in endemic areas.

## 1. Introduction

Leishmaniasis is a neglected tropical disease (NTD) that persists as a major public health concern, closely associated with poverty, affecting millions of people worldwide [1]. American tegumentary leishmaniasis (ATL) is particularly relevant in Latin America, causing cutaneous and mucocutaneous lesions that may lead to disfigurement, disability, and social stigma [2]. The absence of safe, effective, and accessible treatments underscores the urgent need for innovative therapeutic approaches.

Leishmaniasis is caused by protozoa of the genus *Leishmania*, which are transmitted to humans by infected sandflies [3]. More than 20 pathogenic species have been identified that produce visceral, cutaneous, or mucocutaneous forms of the disease [1]. In Argentina, ATL predominates in the northern provinces, where *Leishmania (Viannia) braziliensis*, *Leishmania (Viannia) guyanensis*, and *Leishmania (Leishmania) amazonensis* are the main etiological agents [4,5,6,7,8]. Subtropical climates and socioeconomic vulnerability favor transmission, making timely diagnosis and treatment challenging. The high frequency of therapeutic failures and adverse events associated with reference drugs underscores the need for alternative strategies that can improve adherence, reduce toxicity, and offer oral administration options.

Available treatments rely mainly on pentavalent antimonials, amphotericin B, and paromomycin, all of which present serious drawbacks. Meglumine antimoniate (Glucantime^®^), still used as first-line therapy in Argentina, causes pancreatitis, cardiotoxicity, and hepatotoxicity, necessitating close monitoring and frequently leading to treatment discontinuation [9,10]. The other reference drug, amphotericin B (AmpB), requires intravenous administration, often during hospitalization, due to its risks of nephrotoxicity, hypokalemia, and myocarditis [11,12]. These limitations compromise patient adherence and significantly increase healthcare costs. Moreover, the emergence of resistant strains and coinfections (e.g., HIV/Leishmania) has exacerbated the therapeutic crisis [13].

The current therapeutic pipeline for ATL remains limited. Although natural products (polyphenols, alkaloids, and terpenoids) have shown promising activity, their development is often impeded by high costs, long timelines, and limited translational potential. In contrast, drug repurposing accelerates the path to clinical use by relying on compounds with established pharmacokinetics and safety profiles [9,14]. Even though most repurposed drugs against *Leishmania* are antimicrobial and antineoplastic [9,12], a growing number of non-traditional pharmacological classes have also been explored as antileishmanial candidates. These include bortezomib (a proteasome inhibitor), dibucaine (a local anesthetic), domperidone (a dopamine D_2_ receptor antagonist), acebutolol (a β_1_-adrenergic receptor blocker), prilocaine (a local anesthetic), phenylephrine (an α_1_-adrenergic agonist), and auranofin (an antirheumatic agent). In addition, several central nervous system (CNS) and psychotropic drugs have attracted attention owing to their unexpected antiparasitic effects, including chlorpromazine (a phenothiazine antipsychotic) and cinnarizine (an H_1_-antihistamine and calcium channel blocker) [15,16,17]. Moreover, other repurposed drugs have been proposed for specific clinical manifestations of leishmaniasis: perphenazine and rifabutin for cutaneous forms, particularly as topical formulations [18], and capecitabine and benfotiamine for visceral leishmaniasis, showing potent inhibition of the parasite and its DNA primase [19]. Recent developments have also highlighted ivermectin, particularly when formulated with polymeric micelles [20], and several sterol C-24 methyltransferase inhibitors [21], which are promising candidates despite the need for further in vivo studies to confirm their safety and efficacy. Regarding the antimicrobial agents, while effective in some cases, this approach raises serious concerns about fueling the global crisis of antimicrobial resistance, a phenomenon extensively documented and considered one of the most urgent threats to public health. The rapid emergence and spread of multidrug-resistant pathogens, largely driven by the overuse and repurposing of antibiotics, has already resulted in millions of deaths worldwide and undermines the effectiveness of standard treatments [22]. Another promising avenue is drug combination therapy, which can enhance efficacy, reduce individual drug dosages, and thereby minimize toxicity and resistance development [23]. For AmpB, identifying synergistic partners is translationally valuable to decrease hospitalizations, reduce toxic side effects, and improve adherence.

When considering new therapeutic candidates for ATL, it is essential to evaluate their activity against multiple parasite stages (promastigotes and intracellular amastigotes), assess their impact on parasite survival within host macrophages, and determine potential synergistic interactions with standard reference drugs such as AmpB. To fully establish the translational relevance of these repositioned drugs, future research must prioritize investigating their precise mechanisms of action and extending testing to clinically significant species, such as *L. (V.) braziliensis*. This comprehensive evaluation is essential for robustly assessing their clinical potential and expediting their path to patient use.

Our group previously reported that extracts from high-altitude white grape pomace, rich in acetylcholinesterase (AChE) inhibitory activity, showed marked antileishmanial effects against *L. (L.) amazonensis* [24]. This strong correlation between AChE inhibition and antileishmanial activity supports the exploration of AChE as a relevant therapeutic target. Unlike most repurposing candidates, the clinically approved AChE inhibitors shown in Figure 1, donepezil hydrochloride (DH), rivastigmine tartrate (RT), and galantamine hydrobromide (GH), are not antimicrobials, thereby avoiding cross-resistance with antibacterial or antifungal therapies. Furthermore, they are widely prescribed to elderly patients with Alzheimer’s disease, with oral administration, well-documented tolerability, and manageable side effects [25,26]. These characteristics make them particularly attractive repurposing candidates.

Building on these observations, we hypothesized that clinically approved orally administered AChE inhibitors could offer a more readily translatable therapeutic option. This study aimed to explore the in vitro antileishmanial activity of AChE inhibitors as potential repositioned therapies for ATL. By integrating assays across parasite stages, combination studies with AmpB, and preliminary exploration of mechanisms of action, we sought to provide a rigorous preclinical foundation for the repurposing of these compounds in leishmaniasis treatment.

## 2. Results

### 2.1. In Vitro Cytotoxicity and Antileishmanial Activity of AChE Inhibitors Against Leishmania (L.) amazonensis Promastigotes

The repurposed acetylcholinesterase inhibitors, donepezil hydrochloride (DH), rivastigmine tartrate (RT), and galantamine hydrobromide (GH), demonstrated clear antileishmanial activity against *Leishmania (L.) amazonensis* promastigotes, with all three compounds showing measurable inhibitory effects (Table 1). Among them, DH consistently emerged as the most effective one, with an IC_50_ of 16.82 μM, notably lower than those of RT (87.78 μM) and GH (176.5 μM), indicating superior potency within this group of repurposed drugs. Cytotoxicity assays in mammalian macrophages revealed a favorable profile for all three inhibitors. DH and RT displayed CC_50_ values above 150 μM (172.51 μM and 151.31 μM, respectively), whereas GH reached 404.86 μM, confirming that concentrations substantially higher than those exerting antileishmanial activity were required to induce adverse effects on host cells. These parameters translated into broad in vitro safety windows, particularly for GH (228.36), followed by DH (155.69) and RT (63.53). Selectivity indices (SI) further highlighted this advantage: DH presented the most favorable balance of efficacy and safety with an SI of 10.25, while RT and GH reached 1.72 and 2.29, respectively.

Collectively, these data identify DH as the most promising candidate among the repurposed compounds, combining superior inhibitory activity with a markedly enhanced safety profile. For comparison, the reference drug amphotericin B (AmpB) achieved a lower IC_50_ value (3.25 μM) but at the expense of host toxicity, with a CC_50_ of only 7.58 μM, yielding a narrow safety window of 4.33. Although its SI value (2.33) exceeded those of RT and GH, it remained substantially lower than that of DH, underscoring the translational advantage of repurposed AChE inhibitors as safer alternatives to the current standard of care.

### 2.2. Potent Synergistic Interactions Between Repurposed AChE Inhibitors and Amphotericin B

Building on the single-drug activity profiles, we investigated whether combining the repurposed AChE inhibitors with AmpB could enhance their inhibitory effects against *L. (L.) amazonensis* promastigotes. The interactions were analyzed using isobologram models (Figure 2), in which the IC_50_ concentrations (μM) for each drug combination at five different ratios (AChE inhibitor:AmpB = 1:0, 1:1, 3:1, 10:1, and 0:1; colored symbols) were plotted with AmpB concentrations on the y-axis and the individual AChE inhibitor concentrations on the *x*-axis. The theoretical additive line (red dashed line) connects the IC_50_ values of each drug as a monotherapy and separates synergistic effects (below the line) from antagonistic effects (above the line). All three repurposed drugs displayed synergistic interactions with AmpB across all tested ratios (Figure 2A–C), as evidenced by the consistent positioning of all experimental data points (colored symbols connecting the colored lines) well below the red dashed additive line for DH (Figure 2A), RT (Figure 2B), and GH (Figure 2C). This graphical pattern indicates that the observed IC_50_ values for all combination ratios were substantially lower than those predicted by a simple additive effect, demonstrating synergistic antiparasitic activity.

To quantitatively confirm these visually apparent synergistic interactions, the sum of the fractional inhibitory concentrations (∑FIC) was calculated for each combination ratio (see Section 4.5). This quantitative metric integrates the concentration reductions observed in the isobolograms into a single numerical value, where ∑FIC ≤ 0.5 indicates synergy between the two drugs. Among them, the DH-AmpB combination was the most potent, with a χ∑FIC of 0.09 (mean of the 10:1, 3:1, and 1:1 ratios), demonstrating strong synergy. Notably, in the equimolar 1:1 combination, the resulting ∑FIC value was 0.07 with an IC_50_ of 0.19 μM for both DH and AmpB, representing a drastic 88-fold reduction compared to the IC_50_ of DH alone (16.82 μM, Table 1) and a 17-fold reduction for AmpB (3.25 μM, Table 1). The RT-AmpB and GH-AmpB combinations also showed strong synergistic profiles, with χ∑FIC values of 0.17 and 0.16, respectively (averaged across the tested ratios). For the equimolar 1:1 combinations, the ∑FIC values were 0.156, with an IC_50_ of 0.49 μM for RT-AmpB and 0.14 with an IC_50_ of 0.45 μM for GH-AmpB, representing 179-fold and 393-fold reductions compared to RT (87.78 μM) and GH (176.5 μM) as monotherapies, respectively (Table 1).

The equimolar 1:1 ratio was selected for subsequent experiments because it achieved robust synergistic efficacy comparable to that of ratios with higher AChE inhibitor concentrations (3:1 and 10:1) while minimizing the total drug load. This combination provides an optimal balance between antiparasitic potency and dose-sparing potential. Collectively, these data demonstrate that the inhibitory effect of each AChE inhibitor was significantly enhanced when combined with AmpB. Throughout the subsequent sections, references to IC_50_ of the combination denote the values obtained at the 1:1 ratio (0.19 μM for DH-AmpB, 0.49 μM for RT-AmpB, and 0.45 μM for GH-AmpB), and 2 × IC_50_ refers to twice these concentrations. To assess whether these potentiated antiparasitic effects were accompanied by increased host cytotoxicity, viability assays were performed in RAW 264.7 macrophages and L929 fibroblasts at IC_50_ and 2 × IC_50_ concentrations of each drug combination. No reductions in cell viability were observed under these conditions, indicating that the synergistic activity detected against *L. (L.) amazonensis* promastigotes was not associated with additional cytotoxic effects on mammalian cells.

### 2.3. Antileishmanial Activity of AChE Inhibitors and Their Combinations with Amphotericin B Against Intracellular Amastigotes of Leishmania (L.) amazonensis

Macrophages were infected with *L. (L.) amazonensis* metacyclic promastigotes and treated with each compound, DH, RT, GH, and AmpB, as well as their 1:1 combinations (DH-AmpB, RT-AmpB, GH-AmpB) at concentrations equivalent to the IC_50_ previously determined for promastigotes. Two complementary readouts were employed: (i) microscopy-based quantification of the percentage of infected macrophages and the mean number of amastigotes per infected cell, and (ii) a rescue assay in which intracellular amastigotes were released after treatment, allowed to transform into promastigotes, and quantified them to estimate the proportion of viable survivors.

When the percentage of infected macrophages was measured (Figure 3A, left panel), the following mean values were observed relative to untreated control (defined as 100%): DH alone reduced the proportion of infected cells to 56% of control, whereas the DH-AmpB combination showed a slightly higher percentage of infected macrophages (65%). AmpB alone produced a modest reduction (50%). RT as monotherapy and RT-AmpB yielded 53% and 45% infected macrophages, respectively; while GH and GH-AmpB resulted in 63% and 54% infected macrophages. Overall, these values parallel the results obtained with promastigotes, indicating that the tested concentrations achieve approximately a twofold reduction in the proportion of infected cells at the intracellular amastigote stage.

The mean number of intracellular amastigotes per infected macrophage (Figure 3A, middle panel) also showed clear reductions with AChE inhibitors: Untreated 9.5, DH 3.3, DH-AmpB 0.4, AmpB 7.4, RT 3.6, RT-AmpB 2.0, GH 3.6, and GH-AmpB 0.6. Although the DH-AmpB combination was associated with a relatively high proportion of infected macrophages (65%), those infected cells contained very few amastigotes on average (0.4 amastigotes/cell). In contrast, AmpB alone showed fewer infected macrophages but a substantially higher parasite load per infected cell (7.4 amastigotes/cell). RT and GH as monotherapies produced intermediate reductions in parasite burden (approximately 3.3–3.6 amastigotes/cell), whereas their combinations with AmpB reduced the burden further (RT-AmpB 2.0, GH-AmpB 0.6). The integrated infection index (Figure 3A, right panel) reflects these combined effects, with the DH-AmpB combination yielding the greatest overall reduction compared to untreated controls, reinforcing its potential as previously observed against promastigotes. Representative micrographs for untreated, AmpB, DH, and DH-AmpB experimental conditions are shown in Figure 3B.

The rescue assay provides a viability-based readout (Figure 3C). Results are expressed as percentage recovery relative to untreated controls (defined as 100%). Using promastigote IC_50_-equivalent concentrations, DH reduced recoverable viable parasites to approximately 18%, AmpB to approximately 45%, and the DH-AmpB combination to approximately 7%. RT alone resulted in approximately 21% recovery, whereas RT-AmpB further reduced viability to approximately 8%. GH and GH-AmpB produced recoveries of approximately 55% and 35%, respectively. Notably, all drug combinations demonstrated statistically significant reductions in viable parasite recovery compared to their respective monotherapies (*p* < 0.001 for DH-AmpB vs. DH, RT-AmpB vs. RT, and GH-AmpB vs. GH), validating the synergistic effects observed in promastigotes and confirming that combination therapy enhances antiparasitic efficacy against intracellular amastigotes. These findings corroborate the microscopy-based infection indices, with the rescue assay confirming that the reduction in parasite burden observed microscopically reflects a corresponding decrease in viable parasites capable of transformation. Notably, DH and RT (both individually and in combination with AmpB) demonstrated superior antiparasitic efficacy compared to GH, as evidenced by their greater reduction in both infection indices and viable parasite recovery.

Collectively, the rescue data demonstrate that the strongest reduction in parasite viability was achieved with DH alone when compared to other monotherapies, and with the DH-AmpB combination when compared to other drug combinations. These results emphasize the superior efficacy of donepezil hydrochloride-based treatments, confirming their ability to markedly reduce intracellular survival and reinforcing their translational potential.

### 2.4. Donepezil Hydrochloride Compromises Promastigote Morphology Through Direct and Synergistic Effects with Amphotericin B

Given these prior observations, which identified donepezil hydrochloride as the most promising candidate among the tested AChE inhibitors and demonstrated its strong synergistic activity with AmpB against both promastigotes and intracellular amastigotes, we investigated the morphological alterations associated with treatment. These experiments were performed on promastigotes at the previously determined IC_50_ concentrations, focusing on the DH-AmpB combination compared to monotherapies and untreated controls.

After treatment and Giemsa staining, promastigotes were classified into two distinct morphological populations: typical forms, characterized by elongated cell bodies and a long flagellum, and atypical forms, displaying alterations in body shape or flagellar structure (Figure 4A). Atypical features included cell rounding, body shrinkage or deformation, and flagellar abnormalities such as shortening, loss, or the presence of two asymmetric flagella suggestive of impaired division. As quantified in Figure 4B, untreated controls were predominantly promastigotes with typical form (approximately 92%), with atypical forms accounting for only approximately 8%. Among treated groups, AmpB alone induced approximately 27% atypical forms (*p* < 0.001 vs. untreated). DH treatment also increased the proportion of atypical parasites to approximately 24% (*p* < 0.001 vs. untreated), yielding values like those observed with AmpB. Remarkably, the DH-AmpB combination produced a striking shift, with atypical forms representing approximately 97% of the population (*p* < 0.001 vs. untreated; *p* < 0.001 vs. monotherapies).

Scanning electron microscopy (SEM) provided further evidence of these treatment-induced alterations (Figure 4C). Untreated parasites displayed well-preserved slender cell bodies, smooth surfaces, and elongated flagella. AmpB-treated promastigotes exhibited characteristic membrane damage, including pore formation. DH-treated parasites displayed roughened surfaces and mild deformation. Strikingly, promastigotes exposed to the DH-AmpB combination showed the most pronounced alterations, including twisted cell bodies and flagella, membrane folding, and pore-like structures. These observations suggest that DH and AmpB act through distinct but complementary mechanisms, producing enhanced structural damage when combined.

Flow cytometry was employed to complement the microscopic observations by quantitatively assessing parasite size (forward scatter, FSC) and internal complexity or granularity (side scatter, SSC), as shown in Figure 4D,E. This analysis revealed distinct promastigote populations that shifted according to the applied treatment. In untreated controls, parasites were predominantly distributed in a diagonal cluster designated as P1, representing approximately 76% of the population, consistent with typical morphology. A minor subpopulation (P2, approximately 21%) was also detected, characterized by higher FSC and SSC values, indicating slightly larger and more granular cells.

Redistribution of events was observed following DH treatment, with a marked reduction in P1 (approximately 26%) and a concomitant expansion of P2 to approximately 69%. This shift parallels the microscopic identification of atypical forms, reflecting increased cell size and structural alterations. In contrast, AmpB-treated parasites maintained a high proportion of P1 (approximately 52%) and did not significantly increase P2 (approximately 2%) but instead gave rise to a distinct subpopulation (P3, approximately 46%) positioned at similar FSC but reduced SSC, consistent with loss of internal complexity due to membrane disruption.

Notably, the flow cytometry profiles reflected the characteristics of both monotherapies when DH was combined with AmpB: P1 was further reduced to approximately 51%, P2 expanded to approximately 42%, and P3 accounted for approximately 7% of the events. This mixed distribution suggests that the DH-AmpB combination does not simply amplify the effect of one drug but integrates their distinct morphological impacts, producing a heterogeneous population of damaged parasites displaying both increased size and granularity alongside reduced internal complexity.

Collectively, the integrated evidence from light microscopy, scanning electron microscopy, and flow cytometry demonstrates that DH alone promotes structural alterations compatible with cell enlargement and surface roughening, whereas AmpB primarily induces loss of complexity through pore formation and membrane damage. Their combination magnifies these effects, significantly increasing the proportion of atypical forms and generating severe disruptions that encompass both morphological phenotypes. Given these profound alterations, we sought to further explore the potential mechanisms of action of donepezil hydrochloride, both alone and in synergy with amphotericin B, to gain deeper insight into how these drugs compromise parasite survival.

### 2.5. AChE-Independent Activity of Donepezil Hydrochloride Against L. (L.) amazonensis

To investigate whether the antileishmanial activity of DH is linked to its canonical role as an AChE inhibitor, we first assessed the potential contribution of extracellular choline availability. Promastigotes were cultured for 48 h in medium supplemented with or without choline, and the IC_50_ of DH was determined under both conditions. As shown in Figure 5A, DH inhibited parasite growth with comparable potency regardless of choline supplementation; no statistically significant differences were observed between conditions. Additionally, the IC_50_ determined for laboratory-grade DH (Sigma-Aldrich) did not differ significantly from that obtained using the pharmaceutical-grade formulation (Endoclar^®^), excluding formulation-related effects.

Next, membrane preparations from untreated promastigotes were assayed for AChE activity using acetylthiocholine (ATC) as substrate in the presence of DTNB. Human red blood cell membrane preparations served as positive controls. As expected, the erythrocyte control displayed robust enzymatic activity (Figure 5B), whereas membrane preparations from *L. (L.) amazonensis* promastigotes showed no detectable AChE activity, even at elevated membrane concentrations. These results demonstrate that the antileishmanial effect of DH cannot be attributed to AChE inhibition. Collectively, these findings indicate that DH exerts its antiparasitic effects through AChE-independent mechanisms, consistent with the morphological alterations and cytometric changes described above.

### 2.6. Donepezil Hydrochloride Does Not Affect Membrane Fluidity but Remains Compatible with Amphotericin B Disruption

To further investigate the potential mechanism of action of DH, we evaluated its impact on parasite membrane fluidity using fluorescence polarization with the hydrophobic probe DPH. Promastigotes were treated with AmpB, DH, or their equimolar combination at the previously determined IC_50_ concentrations, followed by membrane purification and DPH labeling. Fluorescence polarization values were measured as indicators of the membrane order. If higher values were observed compared to the control samples, it was concluded that the treatment induced more rigid and ordered membranes. In contrast, if lower values are obtained upon treatment with the drugs, an increase in membrane fluidity should be considered. As shown in Figure 6, untreated promastigote membranes exhibited the highest fluorescence polarization values, consistent with their characteristic ordered lipid environment. AmpB treatment significantly decreased polarization (*p* < 0.01 vs. untreated), in line with its established ability to bind ergosterol and disrupt membrane architecture. In contrast, DH alone yielded values comparable to untreated controls, indicating that it does not directly alter membrane fluidity under these experimental conditions. Notably, parasites treated with the DH-AmpB combination displayed a significant reduction in polarization (*p* < 0.05 vs. untreated), although the effect was less pronounced than with AmpB monotherapy. This difference can be attributed to the lower AmpB concentration in the combination (0.19 µM vs. 3.2 µM as monotherapy), suggesting that the observed depolarization primarily reflects AmpB activity, with DH contributing through alternative mechanisms that neither interfere with nor enhance its membrane-disruptive effects.

### 2.7. Antileishmanial Activity of Donepezil Hydrochloride and Amphotericin B Against the Predominant Regional Species Leishmania (V.) braziliensis

Given the clinical and epidemiological relevance of *Leishmania (V.) braziliensis* in the endemic areas of northwestern Argentina, we assessed the activity of DH, AmpB, and their combination against this species. Promastigotes and axenic amastigotes were evaluated under the same experimental conditions to capture potential stage-dependent effects.

Parasites were treated with the IC_50_ concentrations previously determined for *L. (L.) amazonensis*, and viability was expressed relative to untreated controls (Figure 7). Representative microscopic images of treated parasites are shown in Figure 7A. AmpB displayed potent activity against both developmental stages, reducing viability to approximately 1% of the control in promastigotes and amastigotes alike (Figure 7B). DH alone exhibited significant inhibitory activity, decreasing viability to approximately 65% in promastigotes (*p* < 0.0001) and approximately 35% in amastigotes (*p* < 0.0001), indicating substantially stronger efficacy against the mammalian stage, nearly twice as potent as against promastigotes. The combination of DH and AmpB demonstrated potent synergy, reducing parasite viability in both stages to nearly 1%, mirroring the efficacy of AmpB monotherapy. Crucially, this result was obtained with a 17-fold reduction in AmpB concentration (0.19 μM in the combination vs. 3.25 μM alone). This finding strongly indicates that DH enables a critical dose-sparing effect, preserving maximal antileishmanial efficacy while drastically lowering the required AmpB exposure.

Collectively, these results underscore the translational significance of DH activity for the treatment of *L. (V.) braziliensis*-induced cutaneous leishmaniasis, particularly its enhanced potency against the clinically relevant amastigote stage. Moreover, the ability of DH to support AmpB activity at markedly lower AmpB doses underscores its potential role in combination regimens aimed at reducing toxicity while preserving therapeutic efficacy.

## 3. Discussion

Leishmaniasis exemplifies the intricate interdependence between human, animal, and environmental health, a relationship central to the One Health framework [27]. Despite the substantial clinical and epidemiological burden these diseases impose, no vaccines are currently available, and treatments have changed little over the past century, remaining toxic, invasive, and often poorly tolerated [9,11,28]. These limitations underscore the urgent need to develop safe, patient-friendly, and regionally adapted therapeutic options as a global health priority. Within this context, drug repurposing has gained momentum as a pragmatic strategy to accelerate therapeutic innovation for neglected tropical diseases [29,30]. Our study embraced this approach by evaluating the antileishmanial activity of three orally available acetylcholinesterase (AChE) inhibitors, donepezil hydrochloride (DH), rivastigmine tartrate (RT), and galantamine hydrobromide (GH), both as monotherapies and in synergistic combinations with amphotericin B (AmpB), against *Leishmania* species relevant to American tegumentary leishmaniasis [5,7,31].

Our results validate the potent antileishmanial activity of AChE inhibitors as repurposed drugs against *Leishmania*. Among these compounds, DH emerged as the most promising candidate, displaying potency superior to several previously reported repurposed oral drugs such as cinnarizine (IC_50_ ≈ 34 μM; SI ≈ 4 [32]), omeprazole (IC_50_ ≈ 50 [33]), and ezetimibe (IC_50_ ≈ 30 [34]). Although these values were obtained under slightly different experimental conditions and, in some cases, with distinct *Leishmania* species, they provide useful context for positioning DH within the spectrum of repurposed compounds with antileishmanial potential. All three AChE inhibitors exhibited selectivity indices (SI) greater than 1, with DH displaying the highest SI value and a remarkably broad safety window, substantially wider than that of AmpB. This distinction is critical for translational development, as it indicates that therapeutically efficacious concentrations can be achieved without compromising host cell viability.

These results resonate with earlier reports on natural products exhibiting dual AChE inhibition and antileishmanial effects [35,36]. In fact, our group previously identified AChE-inhibiting natural products from grape pomace with antileishmanial activity against promastigotes [24]. The present findings expand this knowledge by demonstrating that clinically approved AChE inhibitor, already in long-term human use, also hold promise as antiparasitic agents with greater translational potential.

In addition to demonstrating the antileishmanial potential of clinically approved AChE inhibitors and positioning them as promising candidates for ATL therapy, one of the main strengths of this work is the strong synergistic interaction observed between AChE inhibitors and AmpB, as reflected by ΣFIC values well below the 0.5 threshold indicative of synergy [29,37]. DH proved the most effective combination partner, reducing the IC_50_ of AmpB by nearly 17-fold. Such robust synergy suggests complementary mechanisms of action, with AmpB disrupting parasite membranes through ergosterol binding [38,39] and DH inducing distinct structural alterations. Flow cytometry confirmed these divergent yet complementary pathways, and rescue assays corroborated that reduced infection indices translate into corresponding decreases in viable parasite recovery. Similar synergistic strategies with AmpB have been explored with other compounds as part of drug repurposing efforts. Another example is tamoxifen, which demonstrated antileishmanial activity but showed only additive or indifferent interactions with AmpB [40]. Novel formulations have also been investigated to reduce AmpB toxicity while maintaining efficacy. A niosomal combination of AmpB with selenium showed positive interactions, decreasing *L. tropica* promastigote proliferation approximately 4-fold compared to simple combinations [41]. Similarly, organic salts and ionic liquid derivatives of AmpB have been engineered to reduce toxicity while preserving potency, demonstrating the potential of these novel preparations as agents against leishmaniasis [42]. These innovative formulations open promising avenues for future research, particularly when combined with synergistic partners such as DH or other AChE inhibitors, potentially achieving dual toxicity reduction through both improved AmpB delivery and dose-sparing synergy. Previous studies have reported additive and synergistic effects between artemisinin and amphotericin B (AmpB) against *Leishmania donovani* promastigotes and intracellular *L. martiniquensis* amastigotes [43,44]. Specifically, the combination of artesunate (an artemisinin derivative) with AmpB demonstrated strong synergy across multiple *Leishmania* species. This interaction was potentially attributed to artesunate enhancing AmpB binding to ergosterol on the parasite membranes [45]. However, our cell cytometry assay suggests a different interaction profile. In our experiments, the DH-AmpB combination profile mimicked that of DH alone rather than the AmpB profile, i.e., an increase in the P2 population at the expense of the P1 population and keeping a marginal P3 population. These findings suggest that while both compounds may act independently, DH may have the leading role in the mechanism of action of the DH-AmpB combination whereas AmpB might act as an enhancer of DH.

Other combination approaches with reference drugs have been explored, although with variable success. Miltefosine combined with nifuratel showed potent synergy against axenic and intracellular amastigotes in murine models of visceral leishmaniasis [46]. However, compared to our approach, this strategy raises concerns about antimicrobial resistance, as nifuratel is derived from nitrofurantoin, an antibiotic whose repurposing could contribute to the global antimicrobial resistance crisis. Additional combinations tested in clinical or preclinical settings include glucantime with pentoxifylline, levamisole, or imiquimod [47,48,49]. Compared to these approaches, the DH-AmpB combination offers distinct advantages: DH is not an antimicrobial agent, thereby avoiding cross-resistance concerns; it demonstrates robust synergy (χΣFIC = 0.09 vs. higher values for other combinations); and it is already approved for long-term use in elderly populations with well-documented tolerability.

Beyond efficacy and safety considerations, elucidating the mechanism underlying DH’s antileishmanial activity is crucial for understanding this drug’s potential. AChE inhibitors have been proposed previously as potential antileishmanial agents due to their ability to reduce choline levels, an essential precursor for the synthesis of phosphatidylcholine, the major phospholipid in parasite membranes [50,51]. However, our mechanistic exploration confirmed that DH does not exert its antileishmanial effects through AChE inhibition. The absence of detectable AChE activity in parasite membranes, coupled with the lack of any effect of choline supplementation on DH potency, effectively rules out cholinergic targets. These findings align with studies demonstrating that *Leishmania* can synthesize phosphatidylcholine through alternative pathways independently of choline incorporation [52]. This metabolic flexibility may explain why choline supplementation did not rescue parasites from DH treatment in our study, suggesting that the antileishmanial mechanism operates through pathways unrelated to choline depletion. A further clue that choline and phospholipid de novo synthesis is not the primary mechanism is the demonstrated activity of DH against both promastigote and amastigote stages. This is significant because, while promastigotes actively build their phospholipid pools via de novo synthesis, amastigotes predominantly scavenge lipids from the host. If DH targeted de novo synthesis, amastigotes would likely be refractory to its effects [53]. Our findings point toward non-cholinergic mechanisms. In this regard, interference with organellar metabolism, nutrient acquisition, and other vital cellular processes have been proposed [25,51]. It is worth noting that studies on related kinetoplastid parasites have suggested different mechanisms. For instance, treatment with pyridostigmine bromide, an AChE inhibitor used clinically for myasthenia gravis, improved outcomes in a murine *Trypanosoma cruzi* infection model, with effects attributed to cholinergic modulation of host immune responses [54,55].

Membrane fluidity assays revealed that DH alone did not affect polarization values, whereas AmpB did, confirming that DH acts through pathways distinct from, yet compatible with, sterol targeting. Further mechanistic insights may emerge from related models; increased AChE activity reported in mice during acute *Trypanosoma cruzi* infection [56] raises the possibility that non-canonical cholinergic mechanisms could contribute indirectly to the AChE inhibitor activity against *Leishmania*. Collectively, these insights highlight the need for targeted mechanistic investigations, such as assessments of mitochondrial function or protein-interaction mapping, to delineate the precise molecular targets of DH in *Leishmania*.

The implications of these findings extend beyond mechanistic interest. By substantially reducing the effective dose of AmpB required for parasite clearance, combination therapy with DH could mitigate the nephrotoxicity and the intravenous infusion-related adverse effects that currently necessitate hospitalization and hinder treatment adherence. This aligns with the broader agenda of repositioning oral drugs to improve treatment feasibility and tolerability in resource-limited endemic settings [9,10]. Compared to other repositioned drugs, the DH-AmpB combination stands out for its robust synergy, translational plausibility, and clinical relevance.

From a methodological perspective, this study demonstrates the value of integrating multiple experimental approaches, viability assays, microscopy, flow cytometry, and membrane biophysics, into a unified analytical framework. This multiparametric strategy not only strengthens confidence in the observed synergistic effects but also provides a methodological template for future evaluations of drug combinations against neglected parasitic diseases. A further distinctive aspect of our work is the comprehensiveness of the experimental design: we assessed promastigotes, intracellular amastigotes, and their survival and recovery potential, and extended the analysis to *Leishmania (V.) braziliensis*, a species of high epidemiological relevance. The observation that DH exhibited even greater activity against axenic amastigotes, the clinically relevant form, underscores the translational importance of these results. Unlike natural products with limited pharmacological characterization, the compounds evaluated here are repurposed drugs with well-established safety profiles, already approved for chronic use in elderly populations and now tested in the context of acute parasitic infection. This combination of documented safety, mechanistic complementarity, and translational relevance positions DH as a particularly promising candidate for further development.

This study has several limitations that must be acknowledged. First, this study employed well-characterized reference strains. The evaluation of recent clinical isolates with potentially diverse genetic backgrounds would further validate the broad applicability of these findings. Second, while axenic amastigotes serve as established models for preliminary screening, complementary studies with intracellular amastigotes in primary macrophages would provide additional validation of the therapeutic potential within the host cell environment. Third, the precise molecular targets of DH in *Leishmania* remain to be fully elucidated, representing an essential next step toward a mechanistic understanding. Addressing these questions will require comprehensive metabolomic and organelle-specific functional studies, as well as experimental models of cutaneous leishmaniasis, to confirm both efficacy and safety profiles in vivo.

The translational roadmap for DH-AmpB combination therapy encompasses several critical stages: (i) confirmation of in vivo efficacy in established murine models of cutaneous leishmaniasis. (ii) Comprehensive pharmacokinetic/pharmacodynamic (PK/PD) studies are aimed at optimizing dosing regimens and determining whether clinically relevant DH concentrations can achieve the observed synergistic effects. (iii) Evaluation in relevant preclinical models, such as ex vivo human skin models or models using recent clinical isolates, to better predict clinical efficacy. (iv) Assessment of potential immunomodulatory effects that may contribute to overall therapeutic outcomes. Given that DH is already approved for chronic use in Alzheimer’s disease, with a well-established safety profile at therapeutic doses (5–10 mg/day orally), its repurposing for acute leishmaniasis treatment may benefit from an expedited regulatory pathway. Nevertheless, the distinct dosing requirements and treatment durations necessary for antiparasitic versus cognitive applications require rigorous evaluation.

## 4. Materials and Methods

### 4.1. Parasites

*Leishmania (Leishmania) amazonensis* (MHOM/BR/73/M2269) and *Leishmania (Viannia) braziliensis* (MHOM/BR/75/M2903) reference strains were initially cultured in USMARU biphasic medium supplemented with sterile proline balanced salts solution (PBSS) and antibiotics (100 U/mL penicillin and 50 µg/mL streptomycin) at 24 °C [24]. After four days, parasites were transferred to RPMI 1640 medium (pH 7.5; Gibco, New York, NY, USA) supplemented with 10% heat-inactivated fetal bovine serum (FBS; Gibco, USA), and incubated at 24 °C. Under these conditions, both species were allowed to grow until the exponential (three days) or stationary (seven days) phase, depending on the assay to be performed. Axenic amastigotes were generated only for *L. (V.) braziliensis* from metacyclic promastigotes adapted to acidified RPMI 1640 medium (pH 5.5) supplemented with 20% FBS and incubated at 24 °C for 48 h. After adaptation, metacyclic promastigotes (5 × 10^6^ cells/mL) were inoculated into fresh medium and incubated at 34 °C with 5% CO_2_ for three days before use, since intracellular amastigote assays are technically challenging for this species [57,58].

### 4.2. Drugs

All stock solutions were prepared using dimethyl sulfoxide (DMSO) (Sigma-Aldrich, St. Louis, MO, USA). Solutions of acetylcholinesterase (AChE) inhibitor drugs were obtained from commercially available pharmaceutical formulations. One tablet of each drug was dissolved in 1 mL of DMSO to prepare the following stock solutions: Endoclar^®^ (Baliarda, CABA, Argentine; donepezil hydrochloride, 5 mg/mL), Numencial^®^ (Teva, CABA, Argentine; galantamine hydrobromide, 4 mg/mL), and Exelon^®^ (Novartis, Barce-lona, Spain; rivastigmine tartrate, 1.5 mg/mL). Amphotericin B (Sigma-Aldrich, St. Louis, MO, USA) was dissolved in DMSO to obtain a 250 mg/mL stock solution. Additionally, analytical-grade donepezil hydrochloride (Donepezil HCl; Sigma-Aldrich, St. Louis, MO, USA) was prepared in DMSO at 5 mg/mL. All stock solutions were further diluted in the appropriate culture medium for each assay, assuming 100% of the labeled active ingredient for concentration calculations.

### 4.3. Cytotoxicity Assay

Drug cytotoxicity was evaluated in the murine macrophage cell line RAW 264.7 (ATCC Tib-71TM) maintained in RPMI 1640 medium (pH 7.5) supplemented with 10% fetal bovine serum (FBS; Gibco, USA) at 37 °C in a 5% CO_2_ atmosphere. Cells were seeded in 96-well plates at a density of 1 × 10^5^ cells per well and incubated for 48 h with decreasing concentrations (twofold serial dilutions) of donepezil hydrochloride (DH), rivastigmine tartrate (RT), galantamine hydrobromide (GH), or amphotericin B (Amp B). Negative control wells contained only culture medium. Cell viability was determined using the WST-1 tetrazolium assay [2-(2-methoxy-4-nitrophenyl)-3-(4-nitrophenyl)-5-(2,4-disulfophenyl)-2H-tetrazolium, monosodium salt]. The amount of formazan produced by viable cells was quantified by measuring absorbance at 570 nm using an Infinite PRO spectrophotometer (TECAN, Männedorf, Switzerland) operated with I-Control software. Cell cytotoxicity (CC) values were calculated from three independent experiments performed in triplicate according to the equation: CC = (treatment OD × 100/negative control OD). The mean 50% cytotoxic concentration (CC_50_) values were determined using GraphPad Prism 8.0 (GraphPad Software, San Diego, CA, USA).

### 4.4. Antileishmanial Activity Screening in Promastigotes

The antileishmanial activity of the AChE inhibitor drugs was evaluated against promastigotes in the exponential growth phase. Parasites were seeded in 96-well plates at a density of 1 × 10^6^ parasites/mL and exposed to decreasing concentrations (twofold serial dilutions) of DH, RT, or GH. AmpB was used as the positive control for antileishmanial activity, while untreated parasites served as the negative control. Starting from the highest tested concentrations, parasite viability was determined using the WST-1 assay as described above. The half-maximal inhibitory concentration (IC_50_) and the 90% inhibitory concentration (IC_90_) values were calculated using GraphPad Prism 8.0 (GraphPad Software, San Diego, CA, USA). The cytotoxic concentration (CC_50_) and IC_50_ values were used to determine the in vitro safety window (CC_50_-IC_50_) and Selectivity Index (SI = CC_50_/IC_50_).

### 4.5. Determination of Drug Interactions

The interactions between AmpB and the AChE inhibitors DH, RT, and GH were assessed in *L. (L.) amazonensis* promastigotes using the checkerboard microdilution method [59,60]. This approach enables the quantitative evaluation of synergistic, additive, or antagonistic effects between two compounds based on their combined inhibitory activity. Briefly, the previously determined IC_50_ values for each drug were used as reference concentrations. Stock solutions of AmpB and each AChE inhibitor were diluted in culture medium, and appropriate volumes of each dilution were combined in 96-well microplates to generate the following fixed ratios with their respective concentrations: DH-AmpB 10:1 (stock solution, 50 µM:5 µM), 3:1 (50 µM:17 µM), and 1:1 (50 µM:50 µM). RT-AmpB: 10:1 (stock solution, 150 µM:15 µM), 3:1 (150 µM:50 µM), and 1:1 (10 µM:10 µM). GH-AmpB: 10:1 (stock solution, 45.6 µM:4.56 µM), 3:1 (45.6 µM: 15.2 µM), and 1:1 (10 µM:10 µM). Cell viability was assessed using the WST-1 assay, and 50 percent inhibitory concentrations (IC_50_) were obtained by nonlinear regression using GraphPad Prism software, as described in Section 4.4. Fractional inhibitory concentrations (FICs) were calculated as [IC_50_ in combination]/[IC_50_ of drug alone], and the sum of FICs (ΣFIC) was determined as ΣFIC = FIC drug A + FIC drug B. The mean ΣFIC (χΣFIC) was calculated, and the combinatory effect was classified according to Odds (2023) [59] as follows: χΣFIC ≤ 0.5 = synergistic, 0.5 < χΣFIC ≤ 1 = additive, 1 < χΣFIC < 2 = indifferent, and χΣFIC ≥ 2 = antagonistic. Isobolograms were constructed by plotting the combination of IC_50_ (µM) of AmpB with the IC_50_ (µM) of each AChE inhibitor tested, as previously reported [61,62,63,64,65]. In addition, the viability of RAW 264.7 macrophages and L929 fibroblasts was evaluated after 48 h using IC_50_ and 2 × IC_50_ concentrations of each drug combination tested on the promastigotes.

### 4.6. Amastigotes Drug Sensitivity Assays

Antileishmanial activity was evaluated using two approaches: (A) intracellular amastigotes and (B) a rescue assay. RAW 264.7 murine macrophages were seeded at a density of 1 × 10^5^ cells per well: (A) on round glass coverslips or (B) directly into 24-well culture plates Nunc (Thermo Fisher Scientific, Waltham, MA, USA) without coverslips and incubated overnight to allow cell attachment. In both assays, cells were infected with *L. (L.) amazonensis* metacyclic promastigotes at a 20:1 parasite/macrophage ratio, overnight in RPMI 1640 + 10% FBS; Gibco at 37 °C in a 5% CO_2_ atmosphere. After infection, cultures were washed with RPMI 1640 to remove non-internalized parasites. Infected macrophages were incubated for 48 h with IC_50_ or 2 × IC_50_ concentrations of each tested drug, alone or in combination. Control wells received culture medium only. After treatment, in assay (A), cells were washed with RPMI 1640, fixed with 4% paraformaldehyde for 12 min, and washed three times with sterile PBS (pH 7.4). Macrophages were then stained with Giemsa, and 200 cells per sample were counted in triplicate using an optical microscope. The percentage of infected macrophages, the number of amastigotes per macrophage, and the infection index (II) were calculated as follows: II = % infected macrophages × average number of amastigotes per infected macrophage. For assay (B), the rescue assay [66], treated infected macrophages were briefly exposed to 0.05% RPMI-SDS for 30 s to achieve controlled lysis and release of amastigotes. Released amastigotes were incubated in RPMI + 20% FBS at 25 °C for 48 h to allow transformation into promastigotes. Rescued promastigotes viability were determined using the WST-1 tetrazolium assay as was described above.

### 4.7. Morphological Study of L. (L.) amazonensis Promastigotes Treated a Combinated Therapy

The effect of the most potent drug combination identified in previous sections on the morphology of L. (L.) amazonensis promastigotes was evaluated using complementary approaches. Promastigotes in the exponential growth phase were treated with the IC_50_ concentration of the selected combination therapy or with each drug alone and incubated for 48 h at 24 °C. Untreated parasites were included as controls. Following treatment, promastigotes were fixed in 4% paraformaldehyde for 12 min. A fraction of the fixed parasites was sent to the Centro Integral de Microscopía Electrónica (CIME, Tucumán, Argentina) for analysis using a Carl Zeiss^®^ Supra 55VP scanning electron microscope (Oberkochen, Germany). The remaining samples were placed on microscope slides, stained with Giemsa solution (1:10) for 10 min at 37 °C, rinsed with distilled water, and examined under a Leica DM500 optical microscope (Mannheim, Germany). Images were acquired at 1000× magnification using a Leica ICC50 HD camera. Morphological evaluation was performed by counting 100 promastigotes per group in the photomicrographs, classifying them as typical or atypical forms. All experiments were performed in triplicate. For quantitative analysis by flow cytometry, promastigotes (5 × 10^6^ cells/mL) were exposed to the IC_50_ treatment for 48 h, harvested, washed three times with PBS, and fixed in chilled 100% methanol. Fixed cells were washed again with PBS and treated with 1 µg/mL RNase A (GenBiotech, Buenos Aires, Argentina) for 1 h at 37 °C before acquisition on a BD FACS Canto II flow cytometer (BD Biosciences, San Jose, CA, USA). Forward scatter (FSC) and side scatter (SSC) parameters were analyzed to evaluate changes in cell size and internal complexity, respectively. Data were processed using FlowJo vX.0.7 software.

### 4.8. Evaluation of AChE Inhibitor Activity Against Leishmania Under Choline-Limited Conditions

To determine whether AChE inhibitors interfere with *Leishmania* metabolism, drug activity was evaluated under choline-limited culture conditions. After four days cultured in USMARU medium, *L. (L.) amazonensis* promastigotes were cultured in DMEM (Gibco Life Technologies, Grand Island, NY, USA), a medium lacking choline, supplemented with 10% FBS. Two conditions were tested: DMEM without supplementation and DMEM supplemented with choline chloride (Sigma-Aldrich, St. Louis, MO, USA) to a final concentration of 20 mg/L, i.e., a concentration high enough to ensure the supply of choline to promastigotes in order to prevent the need of acetylcholinesterase activity for providing choline for phospholipid synthesis. Parasites in the exponential growth phase were seeded in 96-well plates at a density of 1 × 10^6^ cells/mL and exposed to twofold serial dilutions of analytical-grade DH for 48 h. Cell viability was assessed using the WST-1 assay, and IC_50_ values were calculated as described above (Section 4.3 and Section 4.4).

### 4.9. Assessment of Acetylcholinesterase Activity in Leishmania Promastigote Membrane

AChE activity in *L. (L.) amazonensis* promastigote membranes was determined using Ellman’s method [67]. Purified membranes were resuspended in phosphate buffer (5 mM, pH 7.4) containing 1 mM EDTA and 0.33 mM DTNB. Samples were preincubated at 37 °C for 2 min, and acetylthiocholine (ATC) was then added as a substrate. Absorbance at 412 nm, resulting from the reaction of released thiocholine with DTNB, was monitored at 37 °C for 3 min. Human red blood cell membranes were used as positive controls.

### 4.10. Evaluation of L. (L.) amazonensis Membrane Order

Promastigotes in the exponential growth phase (1 × 10^6^ parasites/mL) were treated for 48 h at 24 °C with the IC_50_ concentration of the most potent drug combination identified or with each drug alone. AmpB at its IC_50_ was used as positive control, while untreated parasites served as negative controls. After incubation, parasites were harvested by centrifugation, lysed by sonication, and centrifuged again to isolate membranes. The fluorophore 1,6-diphenyl-1,3,5-hexatriene (DPH; 10 nM) was added to each membrane preparation in the dark. Fluorescence was measured using an ISS PC1 spectrofluorometer (excitation: 350 nm, emission: 450 nm). Fluorescence polarization was calculated according to: *p* = (F_vv_ − F_vh_)/(F_vv_ + F_vh_), where I_vv_ is the fluorescence intensity measured parallel to the excitation beam polarizer, whereas I_vh_ is the fluorescence intensity measured perpendicular to the excitation light. A decrease in fluorescence polarization compared with untreated controls indicates a less ordered membrane, i.e., an increase in the membrane fluidity.

### 4.11. Effect of Combination Therapy on L. (V.) braziliensis

Promastigotes (1 × 10^6^ parasites/mL) and axenic amastigotes (Section 4.1) were treated with IC_50_ or 2 × IC_50_ of the most potent combination for 48 h at 24 °C. Viability was measured using WST-1 (Section 4.3). Treated cells were fixed, Giemsa-stained, and examined under a Leica DM500 microscope at 1000× using a Leica ICC50 HD camera (Leica Microsystems, Wetzlar, Germany).

### 4.12. Statistical Analysis

Data are presented as mean ± standard error of the mean (SEM) from at least three independent experiments performed in triplicate. Statistical comparisons between groups were performed using one-way analysis of variance (ANOVA) followed by Tukey’s post hoc test. Statistical significance was set at *p* < 0.05. All statistical analyses were performed using GraphPad Prism software (version 8.0).

## 5. Conclusions

This work provides the first demonstration that AChE inhibitor drugs, particularly donepezil hydrochloride, display strong antileishmanial activity against *L*. (*L*.) *amazonensis* both as monotherapy and in synergistic combination with amphotericin B. Among the compounds tested, DH exhibited the most favorable selectivity and safety profile and profoundly enhanced AmpB efficacy across parasite developmental stages. Mechanistic analyses suggest that DH acts through non-cholinergic pathways, complementing rather than overlapping with AmpB’s ergosterol-targeting mechanism. The DH-AmpB combination markedly reduced parasite viability, induced characteristic morphological alterations, and retained maximal efficacy against *L*. (*V*.) *braziliensis*, one of the predominant species causing tegumentary leishmaniasis in South America, particularly in Argentina. These findings are especially novel in representing the first evaluation of clinically approved AChE inhibitors, already widely used and well-tolerated in long-term treatments, as acute antileishmanial agents. By demonstrating robust synergy with AmpB, our study supports their repositioning as viable combination partners to reduce toxic dosing requirements and improve treatment feasibility in endemic regions. Collectively, these results pave the way toward more effective and patient-friendly therapeutic regimens for leishmaniasis, with the potential to improve adherence and reduce the burden of adverse effects associated with current standard treatments.

## Figures and Tables

**Figure 1 antibiotics-14-01182-f001:**
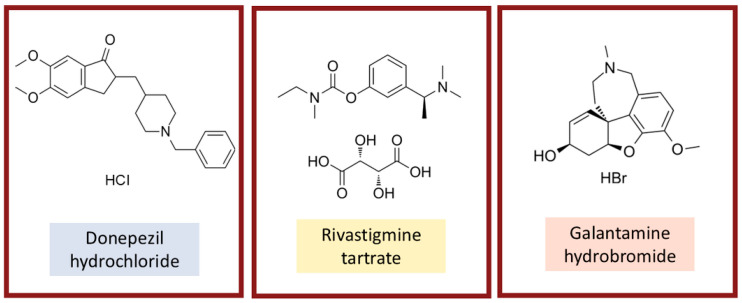
Chemical structures of the acetylcholinesterase inhibitors evaluated in this study: donepezil hydrochloride, rivastigmine tartrate, and galantamine hydrobromide.

**Figure 2 antibiotics-14-01182-f002:**
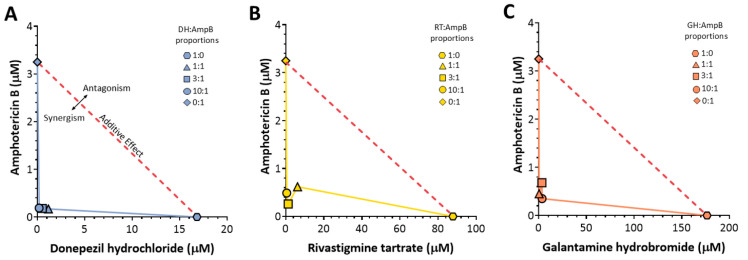
Isobologram analysis of synergistic interactions between amphotericin B and AChE inhibitors against *L. (L.) amazonensis* promastigotes. Each plotted data point signifies the concentration required for 50% inhibition (${\mathrm{IC}}_{50}$), either when applied alone or in a combination (AChE inhibitor: AmpB). The endpoints of the red dashed line correspond to the IC_50_ values of each single drug, and this line represents the theoretical additive effect. Combinations yielding IC_50_ values that fall below this line indicate synergistic interactions, whereas those above indicate antagonistic interactions. (**A**) Donepezil hydrochloride (DH). The blue line connects the IC_50_ experimental data points (blue symbols) representing drug combinations at different ratios (1:0 -⬢- DH alone, 1:1 -▲-,3:1 -■-,10:1 -●-, and 0:1 -◆- AmpB alone). (**B**) Rivastigmine tartrate (RT). The yellow line connects the experimental IC_50_ data points (yellow symbols) at the same combination ratios: 1:0 -⬢- RT alone, 1:1 -▲-,3:1 -■-,10:1 -●-, and 0:1 -◆- AmpB alone). (**C**) Galantamine hydrobromide (GH). The orange line connects the experimental IC_50_ data points (orange symbols) at ratios of 1:0 -⬢- GH alone, 1:1 -▲-,3:1 -■-,10:1 -●-, and 0:1 -◆- AmpB alone. The symbol shapes and their corresponding ratios are indicated in the legend (upper right corner of each panel). The preparation of stock solutions is described in Section 4.5 and the experiments were performed in triplicate.

**Figure 3 antibiotics-14-01182-f003:**
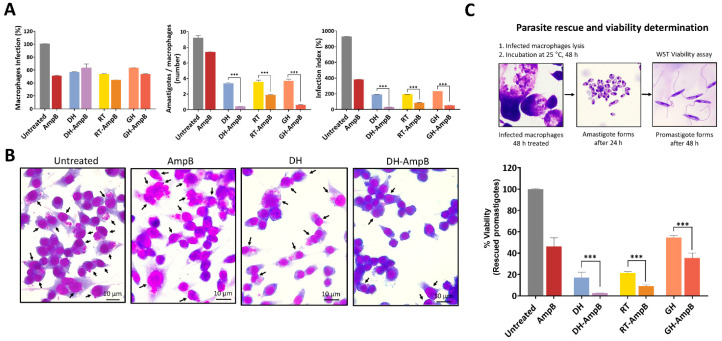
Inhibitory activity of AChE inhibitors and their combinations with amphotericin B against *L. (L.) amazonensis* intracellular amastigotes. (**A**) Left panel: Percentage of infected macrophages after treatment with each compound at concentrations equivalent to the promastigote IC_50_. Values are expressed relative to untreated controls (100%). Middle panel: Mean number of amastigotes per infected macrophage under the same treatment conditions. Right panel: Infection index, calculated as the product of the percentage of infected macrophages and the mean number of amastigotes per infected cell. (**B**) Representative micrographs of untreated and treated macrophages (scale bar = 10 μm). Black arrows indicate infected macrophages. (**C**) Rescue assay showing the percentage of viable amastigotes able to transform into promastigotes following treatment, expressed relative to untreated controls (100%). Schematic representation of the intracellular parasite rescue assay is shown in the upper panel. Data represent mean ± SEM of three independent experiments. Statistical comparisons between groups are indicated: *** *p* < 0.001.

**Figure 4 antibiotics-14-01182-f004:**
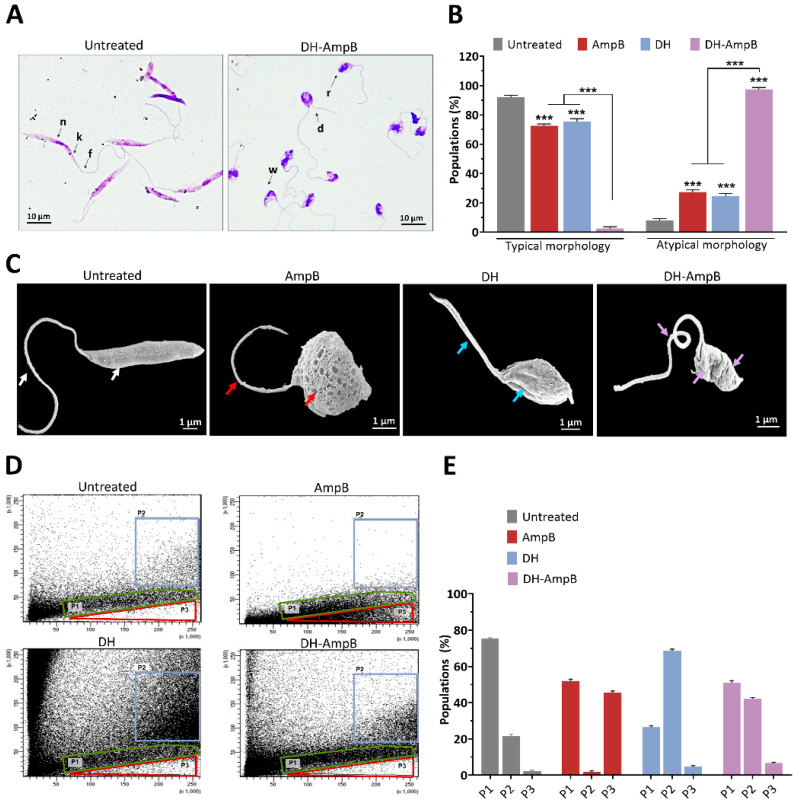
Morphological alterations in *L. (L.) amazonensis* promastigotes treated with donepezil hydrochloride, amphotericin B, or their combination. (**A**) Representative light microscopy images of Giemsa-stained promastigotes (scale bar = 10 μm). Left: untreated promastigotes showing typical morphology; nucleus (n), kinetoplast (k), and flagellum (f) indicated by black arrows. Right: DH-AmpB-treated promastigotes exhibiting atypical morphology, including altered cell body and/or flagellum length, cell rounding (r), deformation of body width (w), and the presence of two asymmetric flagella (d). (**B**) Quantitative analysis of the proportion of typical versus atypical forms after treatment. Values are shown as percentage of the total counted population. (**C**) Scanning electron micrographs (scale bar = 1 μm). The arrows indicate the characteristic structures. (**D**) Flow cytometry dot plots (FSC vs. SSC) identifying distinct parasite subpopulations: P1 (typical) shown in green, P2 (larger and more granular) in blue, and P3 (reduced internal complexity) in red. (**E**) Percentage distribution of cytometry populations (P1 + P2 + P3 = 100%). All treatments were tested at concentrations equivalent to their promastigote IC_50_. Donepezil hydrochloride (DH, 16.82 μM, blue bars); amphotericin B (AmpB, 3.25 μM, red bars); equimolar donepezil hydrochloride and amphotericin B (DH-AmpB, 0.19 μM each, violet bars). Data represent mean ± SEM of three independent experiments. Statistical comparisons versus untreated controls or between groups are indicated: *** *p* < 0.001.

**Figure 5 antibiotics-14-01182-f005:**
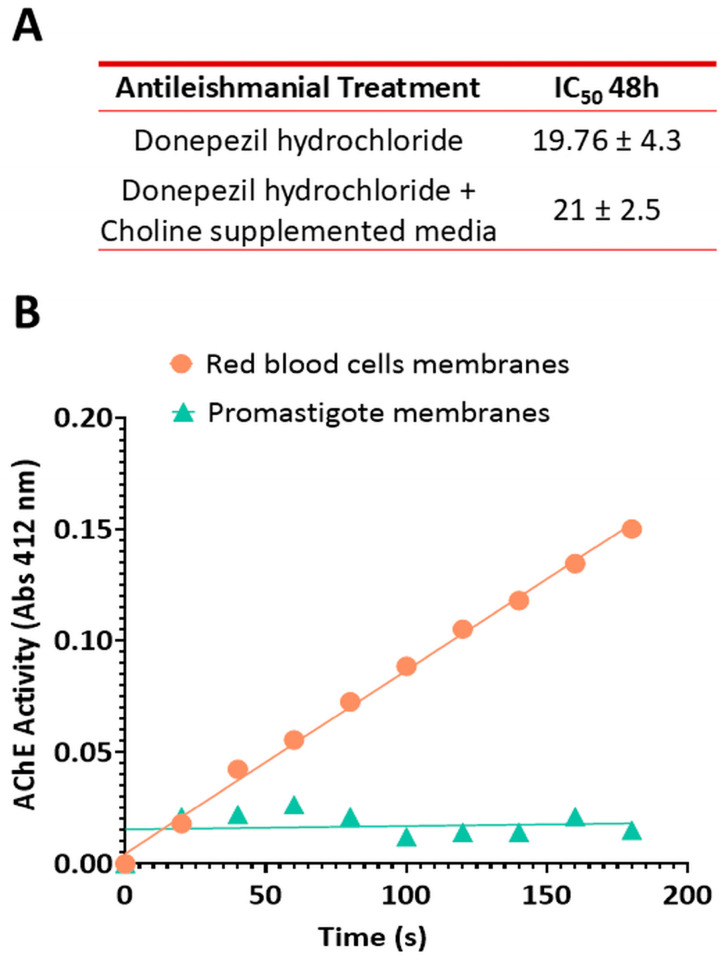
Acetylcholinesterase-independent activity of donepezil hydrochloride against *L. (L.) amazonensis*. (**A**) IC_50_ values of DH in promastigotes cultured for 48 h in medium with or without choline supplementation. Values are expressed in µM (mean ± SEM, *n* = 3). (**B**) Acetylcholinesterase activity assay in membrane preparations from untreated *L. (L.) amazonensis* promastigotes. Acetylthiocholine (ATC) in the presence of 5,5′-dithiobis(2-nitrobenzoic acid) (DTNB) was used as substrate. Human red blood cell membranes served as the positive control and are represented by orange circles. Promastigote membrane preparations are represented by green triangles. Data are representative of three independent experiments.

**Figure 6 antibiotics-14-01182-f006:**
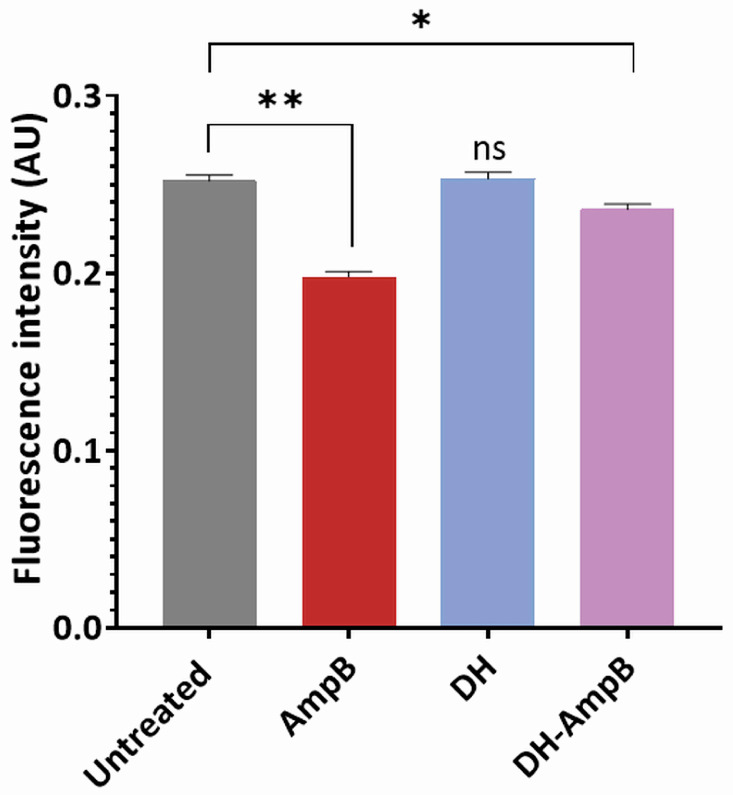
Effect of donepezil hydrochloride and amphotericin B on membrane fluidity of *L. (L.) amazonensis*. Fluorescence polarization of DPH-labeled promastigote membranes after treatment with donepezil hydrochloride (DH, 16.8 µM, blue bar), amphotericin B (AmpB, 3.2 µM, red bar), or the equimolar combination (DH-AmpB, 0.19 µM each, violet bar). Untreated membranes served as control (gray bar). Data represent mean ± SEM of three independent experiments. Statistical comparisons versus untreated controls are indicated: * *p* < 0.05; ** *p* < 0.01; ns, not significant.

**Figure 7 antibiotics-14-01182-f007:**
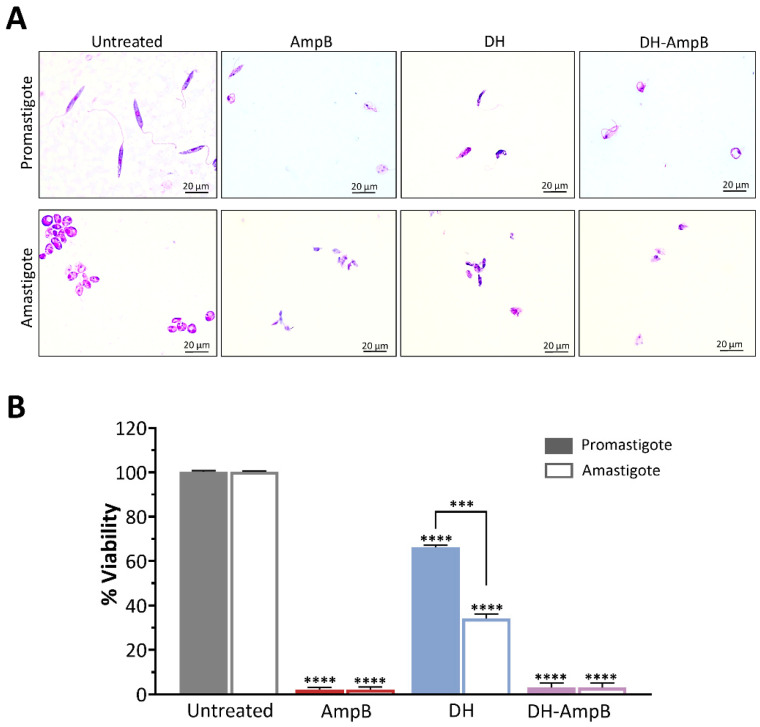
Antileishmanial activity of DH, AmpB, and their combination against *Leishmania (V.) braziliensis*. (**A**) Representative microscopic images of *Leishmania (V.) braziliensis* M2903 promastigotes (upper panels) and axenic amastigotes (lower panels) after 48 h treatment. Scale bars: 20 µm. (**B**) Quantitative assessment of parasite viability for promastigotes (filled bars) and axenic amastigotes (empty bars). Untreated parasites (gray bars), AmpB (3.25 µM, red bars), DH (16.82 µM, blue bars), or the equimolar combination (DH-AmpB, 0.19 µM each, violet bar). Data are expressed as mean ± SEM of three independent experiments. Statistical significance relative to untreated controls is indicated: *** *p* < 0.001, **** *p* < 0.0001.

**Table 1 antibiotics-14-01182-t001:** Antileishmanial activity against *L. (L.) amazonensis* promastigotes, cytotoxicity on RAW 264.7 macrophages, in vitro safety window, and selectivity indices of AChE inhibitors. Values are expressed as μM ± SEM.

Drugs	Promastigote		Macrophage	In Vitro Safety Window	Selectivity Index
	IC_50_ µM	IC_90_ µM	CC_50_ µM		
Donepezil hydrochloride	16.82 ± 1.68	390.38	172.51 ± 18.87	155.69	10.25
Rivastigmine Tartrate	87.78 ± 2.36	437.32	151.31 ± 14.8	63.53	1.72
Galantamine hydrobromide	176.5 ± 5.6	630	404.86 ± 41.9	228.36	2.29
Amphotericin B	3.25 ± 0.5	13.89	7.58 ± 0.6	4.33	2.33

## Data Availability

The original contributions presented in this study are included in the article. Further inquiries can be directed to the corresponding author(s).

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
