# Peer review of "Repurposing Acetylcholinesterase Inhibitors for Leishmaniasis: Donepezil Hydrochloride and Related Compounds Against the American Tegumentary Form"

_antibiotics, 2025, doi:10.3390/antibiotics14121182_

Round 1
Reviewer 1 Report
Comments and Suggestions for Authors
- Why only DH, RT, and GH among many AChE inhibitors were selected for repurposing?
- The introduction does a good job of explaining the idea behind repurposing, but it could be better if it said more clearly why AChE inhibitors were chosen over other non-antimicrobial medications.
- Discuss the translational feasibility of the in vitro ICâ‚…â‚€ of donepezil (≈16.8 µM), which is higher than reported plasma levels in humans (~0.1–0.2 µM) and indicate whether tissue-level accumulation or synergy with amphotericin B could account for this discrepancy.
- Use either “donepezil hydrochloride (DH)” or “donepezil” uniformly throughout.
Author Response
Response to Reviewer 1 Comments
Thank you very much for taking the time to review this manuscript. Please find the detailed responses below and the corresponding revisions/corrections highlighted/in track changes in the re-submitted files.
- Why only DH, RT, and GH among many AChE inhibitors were selected for repurposing?
Even though there are a number of acetylcholine inhibitors, we decided to focus only on those associated with the treatment of Alzheimer’s disease (donepezil, rivastigmine and galantamine) because a side project of the group deals with this pathology. In fact, the present work is based on the results obtained in a previous study of ours where we characterized the antileishmania activity of phenolic compounds that we characterized as AChE inhibitors, in the context of Alzheimer’s. These drugs were selected for this study because they are commercially available in pharmacies and are approved for clinical use by both the U.S. Food and Drug Administration (FDA) and the Argentine National Administration of Medicines, Food and Medical Technology (ANMAT). These acetylcholinesterase (AChE) inhibitors are administered orally (with RT also available as a transdermal patch), in contrast to other AChE inhibitors such as neostigmine, pyridostigmine, physostigmine, or edrophonium, which are given by parenteral routes (intramuscular or intravenous) for specific indications such as myasthenia gravis or reversal of neuromuscular blockade. This distinction is particularly relevant to our work, as one of our main objectives was to explore an orally administered alternative for leishmaniasis treatment, given that the currently available therapeutic options are injectable, require prolonged administration, and are often associated with significant pain and discomfort for patients. Furthermore, they are widely prescribed to elderly patients with Alzheimer's disease, with well-documented tolerability and manageable side effects. Anyway, Neostigmine and pyridostigmine, currently used for treating myasthenia gravis, could be good options for a follow up paper where antileishmania/anti-AChE activities-structure relationship is assessed. To clarify this point, an additional explanation was introduced in lines 137 to 147.
- The introduction does a good job of explaining the idea behind repurposing, but it could be better if it said more clearly why AChE inhibitors were chosen over other non-antimicrobial medications.
This is an interesting point. We directed our research toward anti-AChE compounds because there was no report describing the antileishmania activity of AChE inhibitors. Now, we mention other types of inhibitors that have been included for repurposing in the introduction of the revised version of the manuscript (lines 85 to 106).
- Discuss the translational feasibility of the in vitro ICâ‚…â‚€ of donepezil (≈16.8 µM), which is higher than reported plasma levels in humans (~0.1–0.2 µM) and indicate whether tissue-level accumulation or synergy with amphotericin B could account for this discrepancy.
The reviewer is right pointing out that there is a significant difference between the ICâ‚…â‚€ and the reported plasma levels. However, as the reviewer also noted, the donepezil concentration needed in the presence of amphotericin B is nearly 90 times lower than that of the inhibitor alone. More precisely, although the translational feasibility of the in vitro ICâ‚…â‚€ value of donepezil (≈16.8 µM) may appear limited, as it exceeds typical plasma concentrations reported in humans (generally below 0.4 µM), donepezil is a highly lipophilic compound with extensive tissue distribution, and local concentrations in tissues such as the liver, lungs, and brain may approach or even exceed the in vitro ICâ‚…â‚€. Importantly, when donepezil was combined with amphotericin B, a clear synergistic effect was observed, with ICâ‚…â‚€ values decreasing markedly to 0.19 µM within or close to the range of plasma concentrations achievable in humans. These findings support the translational relevance of the combination. Taking this comment into account, we have discussed this point and emphasized the need for in vivo studies to confirm whether these synergistic effects and pharmacokinetic interactions translate into effective therapeutic outcomes in animal models of leishmaniasis (lines 728 to 744).
- Use either “donepezil hydrochloride (DH)” or “donepezil” uniformly throughout.
Thank you for this comment, we have made the necessary changes to unify donepezil hydrochloride, according to its abbreviation DH, throughout the document, even in the title.

Reviewer 2 Report
Comments and Suggestions for Authors
The present study aimed to investigate the Repurposing Acetylcholinesterase Inhibitors for Leishmaniasis: Donepezil and Related Compounds Against the American Tegumentary Form. Although the work offers some insight into the contact history of neglected tropical diseases, substantial methodological and presentational flaws make the results difficult to interpret.
- Introduction
The introduction highlights a limited number of studies on the current hypothesis of drug repurposing. A more comprehensive review of the relevant literature is required to contextualize the research question adequately.
- Materials and Methods
The description of the combination procedures and the selected ratios is overly brief, and poor study design. Essential details—such as drug/compound information, selected combination ratio protocols, and references must be justified to ensure reproducibility.
- Results
The results are presented with insufficient details; line 205 different ratios were hard to understand, preventing readers from assessing how the data generated the stated conclusions. IC90 of DH is nearly 23 times the high dose (maybe toxic) compared to Amphotericin B (standard drug, nearly 4.2 times). Figure 2. All isobolograms were plotted wrong, with insufficient description. A thorough, figure-by-figure description of the findings is necessary before the study can be critically evaluated. Line 229, non-significant and irrelevant when it says potent synergistic combinations,
- Addressing the above issues is essential before the manuscript can be considered suitable for peer-reviewed publication.
Author Response
Response to Reviewer 2 Comments
Thank you very much for taking the time to review this manuscript. Please find the detailed responses below and the corresponding revisions/corrections highlighted/in track changes in the re-submitted files.
Reviewer #2 Comments:
The present study aimed to investigate the Repurposing Acetylcholinesterase Inhibitors for Leishmaniasis: Donepezil and Related Compounds Against the American Tegumentary Form. Although the work offers some insight into the contact history of neglected tropical diseases, substantial methodological and presentational flaws make the results difficult to interpret.
- Introduction
-The introduction highlights a limited number of studies on the current hypothesis of drug repurposing. A more comprehensive review of the relevant literature is required to contextualize the research question adequately.
We appreciate this observation and understand the importance of the reviewer pointed out. We tried to keep the introduction as concise as possible, mentioning what we considered representative examples of drugs repositioned for treating leishmaniasis, but we agree that a more comprehensive review may improve the manuscript, hence we added more information on the matter, mainly based on what was described in three reviews, which we found quite interesting: Oualha et al. (2024) Front. Cell. Infect. Microbiol. 14: 1403589, Scheiffer et al (2024) EXCLI J 23:1117–1169 and Charlton et al. (2018) Parasitology 145: 219–236. Besides, we found recent works related to repurposed drugs for leishmaniasis such as Freitas et al. (2023) Cytokine 164: 156143, Kumari et al. (2024) Acta Tropica 258: 107338, Nath et al. (2024) Scientific Reports 14: 3246, Bustamante et al. (2019) Journal of Computer-Aided Molecular Design 33: 845–854. Tabrez et al. (2021) Drug Development Research 82, 1154–116. Please see lines 84 to 107.
- Materials and Methods
-The description of the combination procedures and the selected ratios is overly brief, and poor study design. Essential details—such as drug/compound information, selected combination ratio protocols, and references must be justified to ensure reproducibility.
We thank the reviewer for their valuable feedback. It is important for us to clarify the wording of this section for better understanding and reproducibility. Regarding information on the drugs used, we have specified this information in sections 4.2 and 4.5 to ensure the reproducibility of our trials. The revised section 4.5 now includes a detailed description of drug preparation, dilution schemes, data analysis, calculation of the fractional inhibitory concentrations, and isobolograms plotting to ensure full reproducibility. Supporting references were added.
- Results
-The results are presented with insufficient details; line 205 different ratios were hard to understand, preventing readers from assessing how the data generated the stated conclusions.
Thank you very much for your valuable feedback. This comment was essential in improving the clarity and understanding of our findings. For this reason, we have revised and corrected the entire explanation in -Section 2.2 (lines 199 to 245). In addition, we have included further details in Section 4.5 (Materials and Methods) to enhance comprehension. We believe these improvements will facilitate a clearer understanding of the assays performed to validate the drug interactions.
-IC90 of DH is nearly 23 times the high dose (maybe toxic) compared to Amphotericin B (standard drug, nearly 4.2 times).
We appreciate your observation. The interpretation of the IC₉₀ value of donepezil hydrochloride (DH) may not be directly comparable to that of amphotericin B (AmpB) in terms of “high dose” or toxicity. The IC₉₀ represents the concentration required to inhibit 90% of parasite growth, not a cytotoxic concentration. A higher IC₉₀ value for DH simply reflects a lower in vitro potency against promastigotes compared to AmpB, but it does not imply toxicity, since cytotoxicity is independently assessed through the CCâ‚…â‚€ value in macrophages. In this study, DH exhibited a CCâ‚…â‚€ of 172.5 ± 18.9 µM and an ICâ‚…â‚€ of 16.8 ± 1.7 µM, resulting in a selectivity index (SI) of 10.25, which is notably higher than that of AmpB (SI = 2.33). Therefore, despite its higher IC₉₀, DH demonstrates a broader in vitro safety window and a more favorable selectivity profile toward the parasite over host cells. Taking this comment into account, we have also discussed and emphasized the need for in vivo studies to confirm whether the synergistic effects and pharmacokinetic interactions translate into effective therapeutic outcomes in animal models of leishmaniasis (lines 728 to 744).
-Figure 2. All isobolograms were plotted wrong, with insufficient description. A thorough, figure-by-figure description of the findings is necessary before the study can be critically evaluated. Line 229, non-significant and irrelevant when it says potent synergistic combinations,
We sincerely thank the reviewer for their valuable and insightful comments. To enhance clarity for readers, we have added a comprehensive explanation of these results in Section 2.2, and incorporated additional methodological details into the Materials and Methods section. Relevant references were also included to strengthen the context. Figure 2 has been modified, and its caption thoroughly revised in accordance with the reviewer’s suggestions.
The isobolograms were generated following the methodology described in the cited literature (Marmol et al., 2019; Simões Silva et al., 2019; Muhammad et al., 2020, among other references added in the revised version). In our analysis, each point represents the ICâ‚…â‚€ (µM) obtained from the combinations of DH and AmpB (10:1, 3:1, and 1:1), RT and AmpB (10:1, 3:1, and 1:1), and GH and AmpB (10:1, 3:1, and 1:1). Although alternative approaches for constructing isobolograms have been reported (Borges et al., 2023; Trinconi et al., 2014; Gogou et al., 2022), we considered that the selected method offers the clearest and most representative visualization of our results. Specifically, plotting the concentrations of the AChE inhibitor on the x-axis and AmpB on the y-axis allows synergistic interactions to be readily observed relative to the individual ICâ‚…â‚€ values of each compound.

Reviewer 3 Report
Comments and Suggestions for Authors
This research is interesting by repurposing of current acetylcholine esterase inhibitors into ATL medication by co-administration with AmpB.
The manuscript was written nicely. The methodology was well designed to answer the research questions and the results obtained is strongly enough to claim the results of mechanism of DH in synergistic with AmpB as anti-ALT agents. The drawbacks of this research was lack of in-vivo and in-silico testing to confirm the in-vitro results, and also needed other tests to find out the indeed mechanism of DH for anti-ALT. However, authors have already mentioned these drawbacks in the article for readers to understand the circumstances and what they will do next.
I have no problem to get this article to be published after this minor corrections.
- Authors should cut the results of current study off the introduction part (Line 132 onwards). The introduction may mention previous findings and aim of the current study only.
- IC50 in line 765 should be "IC50".
- In methodology section 4.8, authors referred cell viability testing to 4.4. It rather be 4.3 or not, please check.
Author Response
Response to Reviewer 3 Comments
Thank you very much for taking the time to review this manuscript. Please find the detailed responses below and the corresponding revisions/corrections highlighted/in track changes in the re-submitted files.
Reviewer #3 comments:
This research is interesting by repurposing of current acetylcholine esterase inhibitors into ATL medication by co-administration with AmpB.
The manuscript was written nicely. The methodology was well designed to answer the research questions and the results obtained is strongly enough to claim the results of mechanism of DH in synergistic with AmpB as anti-ALT agents. The drawbacks of this research was lack of in-vivo and in-silico testing to confirm the in-vitro results, and also needed other tests to find out the indeed mechanism of DH for anti-ALT. However, authors have already mentioned these drawbacks in the article for readers to understand the circumstances and what they will do next.
I have no problem to get this article to be published after this minor corrections.
- Authors should cut the results of current study off the introduction part (Line 132 onwards). The introduction may mention previous findings and aim of the current study only.
Thank you very much for this comment; our primary intention was to summarize the main results of the manuscript but the reviewer is right about this point. We followed the recommendation and finished the introduction as the reviewer suggested (see lines 145 to 158).
- IC50 in line 765 should be "IC50".
The error was fixed in the revised manuscript.
- In methodology section 4.8, authors referred cell viability testing to 4.4. It rather be 4.3 or not, please check.
We want to thank the reviewer for bringing this up. The viability assay is presented in section 4.3, although macrophages are the focus of the analysis. We described promastigotes viability assay in section 4.4 and that is why we referred to this section when viability was tested in the presence and absence of choline (section 4.8). We corrected this issue in the revised version of the manuscript by changing the last sentence of the section 4.8 for a more general statement that includes both section 4.3 and section 4.4: “Cell viability was assessed using the WST-1 assay, and ICâ‚…â‚€ values were calculated as described above (Sections 4.3 and 4.4)”.

Reviewer 4 Report
Comments and Suggestions for Authors
Summary and Manuscript strengths
The authors present a comprehensive and well-executed study on the repurposing of acetylcholinesterase (AChE) inhibitors—donepezil hydrochloride (DH), rivastigmine tartrate (RT), and galantamine hydrobromide (GH)—as potential therapeutic agents for American tegumentary leishmaniasis (ATL). This work addresses a critical gap in the treatment landscape of a neglected tropical disease by evaluating clinically approved drugs with well-established safety profiles. The manuscript is particularly noteworthy for its comprehensive experimental design, which includes evaluation across multiple parasite stages (promastigotes and intracellular amastigotes), testing against clinically relevant species (L. amazonensis and L. braziliensis), and thorough mechanistic investigations including morphological analysis, flow cytometry, membrane fluidity assays, and enzyme activity assessments. The authors should be commended for their systematic approach to drug combination studies with amphotericin B, demonstrating robust synergistic effects that could translate to dose-sparing regimens with reduced toxicity.
The research is methodologically sound and demonstrates excellent integration of multiple complementary techniques to provide a holistic understanding of the antileishmanial effects. Among the three AChE inhibitors tested, donepezil hydrochloride emerged as the most promising candidate with superior potency (ICâ‚…â‚€ = 16.82 μM against promastigotes) and an impressive selectivity index of 10.25. The demonstration of strong synergy with amphotericin B (χΣFIC = 0.09 for DH-AmpB), reducing the ICâ‚…â‚€ of AmpB by nearly 90-fold, represents a significant finding with substantial translational potential. The authors' mechanistic investigations revealing that DH acts through non-cholinergic pathways add considerable scientific value, distinguishing this work from previous studies on AChE inhibitors. The manuscript is well-written, clearly structured, and appropriately referenced, making it accessible to both specialists and general readers interested in drug repurposing for neglected diseases.
The authors have successfully positioned their findings within the broader context of antimicrobial resistance and the One Health framework, emphasizing that AChE inhibitors, unlike most repurposed antimicrobials, do not contribute to cross-resistance concerns. The comprehensive evaluation extending to L. (V.) braziliensis, a predominant species in South American ATL, strengthens the clinical relevance of these findings. Overall, this manuscript represents a valuable contribution to the leishmaniasis research field and provides a strong foundation for advancing these compounds toward preclinical development.
Specific Comments and Suggestions for Improvement
- Clarification of Drug Formulation Source and Quality Control
Lines 663-672 describe the preparation of stock solutions from "commercially available pharmaceutical formulations" (one tablet dissolved in DMSO). While the authors later compare pharmaceutical-grade with analytical-grade donepezil it would be valuable to provide more specific details about the pharmaceutical formulations used for RT and GH similar to how Endoclar® is mentioned for DH.
This would strengthen the methodology to clarify whether purity analysis or verification of active ingredient content was performed for the tablets, or whether the calculations assumed 100% of the labeled active ingredient. This information would aid reproducibility and help readers assess potential contributions from excipients.
- Statistical Analysis Description
While the manuscript presents data as mean ± SEM from three independent experiments (mentioned throughout Results section), a comprehensive description of statistical methods is notably absent from the Materials and Methods section.
Suggestion: Please add a subsection in Materials and Methods (section 4) titled "Statistical Analysis" that describes the specific tests used. For example: "Data are presented as mean ± standard error of the mean (SEM) from at least three independent experiments performed in triplicate. Statistical comparisons between groups were performed using [one-way ANOVA followed by Tukey's post-hoc test / unpaired Student's t-test / appropriate test]. P-values <0.05 were considered statistically significant. All statistical analyses were performed using GraphPad Prism 8.0 software." This addition would enhance methodological transparency and reproducibility.
- Choline Supplementation Experiment Details
The choline supplementation experiment (lines 772-783, results in lines 384-394) is a crucial control for ruling out AChE-dependent mechanisms. However, some methodological details could be clarified.
Suggestion: Please clarify the rationale for selecting 20 mg/L as the choline supplementation concentration. Was this based on physiological concentrations, literature precedent, or preliminary optimization? Additionally, it would be helpful to specify whether parasites were pre-adapted to choline-free DMEM before drug treatment, or whether the 48-hour drug exposure period also served as the adaptation period. For example: "Parasites were cultured in choline-free DMEM for [X hours] prior to drug addition to ensure adaptation, or parasites were directly transferred to choline-supplemented/unsupplemented medium at the time of drug treatment."
This clarification would aid in interpreting whether choline availability throughout the entire culture period versus during drug treatment specifically was being tested.
- Comparative Analysis with Other Repurposed Drugs
The authors provide excellent context comparing DH with other repurposed drugs (lines 507-514, Discussion), but this comparison could be expanded slightly to better position the findings.
Suggestion: Consider adding specific ICâ‚…â‚€ and SI values for the comparison drugs to provide readers with concrete benchmarks. For example: "DH (ICâ‚…â‚€ = 16.82 μM, SI = 10.25) demonstrated superior potency compared to cinnarizine (ICâ‚…â‚€ = ~XX μM, SI = ~XX [ref 25]), omeprazole (ICâ‚…â‚€ = ~XX μM [ref 26]), and ezetimibe (ICâ‚…â‚€ = ~XX μM, SI = ~XX [ref 27])."
If these specific values are available in the cited references, including them would provide readers with clearer context for evaluating the relative promise of DH. If the cited papers used different Leishmania species or experimental conditions that preclude direct comparison, a brief acknowledgment of this caveat would be appropriate.
- Limitations Section Enhancement
The authors acknowledge limitations (lines 638-647) in a transparent and appropriate manner. However, this section could be slightly expanded to address a few additional considerations.
Suggestion: Consider adding brief mention of: (1) the use of reference strains rather than recent clinical isolates, which may have different drug susceptibility profiles; (2) the use of pharmaceutical formulations (tablets) as drug sources, though this was appropriately validated; and (3) the limitation that axenic amastigotes, while validated models, may not fully recapitulate the intracellular environment. For example: "Additionally, this study employed well-characterized reference strains; evaluation of recent clinical isolates with potentially diverse genetic backgrounds would further validate the broad applicability of these findings. While axenic amastigotes serve as established models for preliminary screening, complementary studies with intracellular amastigotes in primary macrophages would provide additional validation of therapeutic potential in the host cell environment."
These additions would demonstrate thoroughness without undermining the significance of the work.
- Future Directions and Translational Pathway
The manuscript would benefit from a brief, explicit discussion of the translational pathway forward, building on the strong foundation presented.
Suggestion: Consider adding a brief paragraph near the end of the Discussion (before or after the current limitations section) outlining logical next steps toward clinical translation. For example: "The translational pathway forward for DH-AmpB combination therapy would logically include: (1) in vivo efficacy studies in established murine models of cutaneous leishmaniasis, (2) pharmacokinetic/pharmacodynamic (PK/PD) studies to optimize dosing regimens and determine whether clinically achievable DH concentrations can produce the observed synergistic effects, (3) evaluation in ex vivo human skin models or recent clinical isolates to better predict clinical efficacy, and (4) assessment of potential immunomodulatory effects that may contribute to therapeutic outcomes. Given that DH is already approved for chronic use in Alzheimer's disease with well-established safety profiles at therapeutic doses (5-10 mg/day orally), repurposing for acute leishmaniasis treatment may benefit from expedited regulatory pathways, though the different dosing requirements and duration for antiparasitic versus cognitive applications must be carefully evaluated."
This addition would provide readers with a clearer vision of how this promising in vitro work might advance towards clinical application.
Author Response
Response to Reviewer 4 Comments
Thank you very much for taking the time to review this manuscript. Please find the detailed responses below and the corresponding revisions/corrections highlighted/in track changes in the re-submitted files.
- Reviewer #4 comments:
Summary and Manuscript strengths
The authors present a comprehensive and well-executed study on the repurposing of acetylcholinesterase (AChE) inhibitors—donepezil hydrochloride (DH), rivastigmine tartrate (RT), and galantamine hydrobromide (GH)—as potential therapeutic agents for American tegumentary leishmaniasis (ATL). This work addresses a critical gap in the treatment landscape of a neglected tropical disease by evaluating clinically approved drugs with well-established safety profiles. The manuscript is particularly noteworthy for its comprehensive experimental design, which includes evaluation across multiple parasite stages (promastigotes and intracellular amastigotes), testing against clinically relevant species (L. amazonensis and L. braziliensis), and thorough mechanistic investigations including morphological analysis, flow cytometry, membrane fluidity assays, and enzyme activity assessments. The authors should be commended for their systematic approach to drug combination studies with amphotericin B, demonstrating robust synergistic effects that could translate to dose-sparing regimens with reduced toxicity.
The research is methodologically sound and demonstrates excellent integration of multiple complementary techniques to provide a holistic understanding of the antileishmanial effects. Among the three AChE inhibitors tested, donepezil hydrochloride emerged as the most promising candidate with superior potency (ICâ‚…â‚€ = 16.82 μM against promastigotes) and an impressive selectivity index of 10.25. The demonstration of strong synergy with amphotericin B (χΣFIC = 0.09 for DH-AmpB), reducing the ICâ‚…â‚€ of AmpB by nearly 90-fold, represents a significant finding with substantial translational potential. The authors' mechanistic investigations revealing that DH acts through non-cholinergic pathways add considerable scientific value, distinguishing this work from previous studies on AChE inhibitors. The manuscript is well-written, clearly structured, and appropriately referenced, making it accessible to both specialists and general readers interested in drug repurposing for neglected diseases.
The authors have successfully positioned their findings within the broader context of antimicrobial resistance and the One Health framework, emphasizing that AChE inhibitors, unlike most repurposed antimicrobials, do not contribute to cross-resistance concerns. The comprehensive evaluation extending to L. (V.) braziliensis, a predominant species in South American ATL, strengthens the clinical relevance of these findings. Overall, this manuscript represents a valuable contribution to the leishmaniasis research field and provides a strong foundation for advancing these compounds toward preclinical development.
- We really appreciate the positive feedback from the reviewer.
Specific Comments and Suggestions for Improvement
- Clarification of Drug Formulation Source and Quality Control
Lines 663-672 describe the preparation of stock solutions from "commercially available pharmaceutical formulations" (one tablet dissolved in DMSO). While the authors later compare pharmaceutical-grade with analytical-grade donepezil it would be valuable to provide more specific details about the pharmaceutical formulations used for RT and GH similar to how Endoclar® is mentioned for DH.
This would strengthen the methodology to clarify whether purity analysis or verification of active ingredient content was performed for the tablets, or whether the calculations assumed 100% of the labeled active ingredient. This information would aid reproducibility and help readers assess potential contributions from excipients.
- We thank the reviewer for their valuable feedback. It is important for us to clarify the wording of this section for better understanding and reproducibility. For this reason, we have modified section 4.2 with more details on drug preparation (lines 775 to 783). Since our main objective was to reposition drugs generally used in the treatment of Alzheimer’s disease, we decided to use the commercially available pharmaceutical formulations for most experiments. However, we used the analytical-grade donepezil hydrochloride in some experiments assessing the IC50 and got reproducible results: 16.82 ± 1.68 µM with the pharmaceutical formulation and 19.76 ± 4.63 µM with the analytical grade drug. We are confident there are no contributions from the excipients based on the good match between these two approaches. We have not run experiments with analytical grade rivastigmine nor galantamine since these two compounds turned out to be less active than donepezil and we believed no further testing was needed.
- Statistical Analysis Description
While the manuscript presents data as mean ± SEM from three independent experiments (mentioned throughout Results section), a comprehensive description of statistical methods is notably absent from the Materials and Methods section.
Suggestion: Please add a subsection in Materials and Methods (section 4) titled "Statistical Analysis" that describes the specific tests used. For example: "Data are presented as mean ± standard error of the mean (SEM) from at least three independent experiments performed in triplicate. Statistical comparisons between groups were performed using [one-way ANOVA followed by Tukey's post-hoc test / unpaired Student's t-test / appropriate test]. P-values <0.05 were considered statistically significant. All statistical analyses were performed using GraphPad Prism 8.0 software." This addition would enhance methodological transparency and reproducibility.
- We want to thank the reviewer for this suggestion. Unfortunately, during the final editing process, that section, which was originally included, was inadvertently omitted. Section 4.12, Statistical Analysis, has now been reinstated in the Materials and Methods to enhance the methodological transparency and reproducibility of our work (lines 954-961).
Choline Supplementation Experiment Details
The choline supplementation experiment (lines 772-783, results in lines 384-394) is a crucial control for ruling out AChE-dependent mechanisms. However, some methodological details could be clarified.
Suggestion: Please clarify the rationale for selecting 20 mg/L as the choline supplementation concentration. Was this based on physiological concentrations, literature precedent, or preliminary optimization? Additionally, it would be helpful to specify whether parasites were pre-adapted to choline-free DMEM before drug treatment, or whether the 48-hour drug exposure period also served as the adaptation period. For example: "Parasites were cultured in choline-free DMEM for [X hours] prior to drug addition to ensure adaptation, or parasites were directly transferred to choline-supplemented/unsupplemented medium at the time of drug treatment."
This clarification would aid in interpreting whether choline availability throughout the entire culture period versus during drug treatment specifically was being tested.
- We want to thank the reviewer for this suggestion, which was included in the revised manuscript. The methodological details regarding the choline supplementation experiment have been clarified in Sections 4.1 and 4.8 of the Materials and Methods. Specifically, we now specify that L. (L.) amazonensis promastigotes were transferred from USMARU medium to DMEM, a choline-free medium supplemented with 10% FBS. We also clarified why choline was added at a concentration of 20 mg/L. These details have been incorporated to improve methodological transparency and ensure a more accurate interpretation of the experimental design and results.
The physiological concentration of choline ranges from 7 to 20 μM. We used 20 mg/L that is equivalent to ~200 μM, i.e. a concentration 10 times higher than the physiological one, a concentration high enough to ensure cells will not need acetylcholinesterase activity in order to provide choline for phospholipid synthesis. We previously ran a number of preliminary tests with different concentrations of choline in order to establish high concentrations with no toxic effects and 200 μM proved to be an adequate concentration for this purpose.
- Comparative Analysis with Other Repurposed Drugs
The authors provide excellent context comparing DH with other repurposed drugs (lines 507-514, Discussion), but this comparison could be expanded slightly to better position the findings.
Suggestion: Consider adding specific ICâ‚…â‚€ and SI values for the comparison drugs to provide readers with concrete benchmarks. For example: "DH (ICâ‚…â‚€ = 16.82 μM, SI = 10.25) demonstrated superior potency compared to cinnarizine (ICâ‚…â‚€ = ~XX μM, SI = ~XX [ref 25]), omeprazole (ICâ‚…â‚€ = ~XX μM [ref 26]), and ezetimibe (ICâ‚…â‚€ = ~XX μM, SI = ~XX [ref 27])."
If these specific values are available in the cited references, including them would provide readers with clearer context for evaluating the relative promise of DH. If the cited papers used different Leishmania species or experimental conditions that preclude direct comparison, a brief acknowledgment of this caveat would be appropriate.
- This is an excellent point and we appreciate this valuable suggestion. In response, we have expanded the comparative discussion in Section 5 to include specific ICâ‚…â‚€ and SI values for the reference drugs, providing a clearer benchmark for DH activity (lines 576 to 582). This addition strengthens the comparative analysis and enhances the interpretative value of our findings.
- Limitations Section Enhancement
The authors acknowledge limitations (lines 638-647) in a transparent and appropriate manner. However, this section could be slightly expanded to address a few additional considerations.
Suggestion: Consider adding brief mention of: (1) the use of reference strains rather than recent clinical isolates, which may have different drug susceptibility profiles; (2) the use of pharmaceutical formulations (tablets) as drug sources, though this was appropriately validated; and (3) the limitation that axenic amastigotes, while validated models, may not fully recapitulate the intracellular environment. For example: "Additionally, this study employed well-characterized reference strains; evaluation of recent clinical isolates with potentially diverse genetic backgrounds would further validate the broad applicability of these findings. While axenic amastigotes serve as established models for preliminary screening, complementary studies with intracellular amastigotes in primary macrophages would provide additional validation of therapeutic potential in the host cell environment."
These additions would demonstrate thoroughness without undermining the significance of the work.
- We really appreciate the suggestion, and we included it in the revised manuscript. Plese see lines 710 to 720.
- Future Directions and Translational Pathway
The manuscript would benefit from a brief, explicit discussion of the translational pathway forward, building on the strong foundation presented.
Suggestion: Consider adding a brief paragraph near the end of the Discussion (before or after the current limitations section) outlining logical next steps toward clinical translation. For example: "The translational pathway forward for DH-AmpB combination therapy would logically include: (1) in vivo efficacy studies in established murine models of cutaneous leishmaniasis, (2) pharmacokinetic/pharmacodynamic (PK/PD) studies to optimize dosing regimens and determine whether clinically achievable DH concentrations can produce the observed synergistic effects, not only sobre la ulcera sino tambien sobre otros tejidos que pueda alcanzar el parásito (3) evaluation in ex vivo human skin models or recent clinical isolates to better predict clinical efficacy, and (4) assessment of potential immunomodulatory effects that may contribute to therapeutic outcomes. Given that DH is already approved for chronic use in Alzheimer's disease with well-established safety profiles at therapeutic doses (5-10 mg/day orally), repurposing for acute leishmaniasis treatment may benefit from expedited regulatory pathways, though the different dosing requirements and duration for antiparasitic versus cognitive applications must be carefully evaluated."
This addition would provide readers with a clearer vision of how this promising in vitro work might advance towards clinical application.
- This is an excellent input, we appreciate your kind suggestion. It was included in the last paragraph of the discussion (lines 729 to 743).

Round 2
Reviewer 2 Report
Comments and Suggestions for Authors
The authors have thoroughly addressed the reviewer’s comments and improved the manuscript. I recommend acceptance.